# Targeting Immune Senescence in Atherosclerosis

**DOI:** 10.3390/ijms232113059

**Published:** 2022-10-27

**Authors:** Danusha Michelle Vellasamy, Sin-Jye Lee, Khang Wen Goh, Bey-Hing Goh, Yin-Quan Tang, Long Chiau Ming, Wei Hsum Yap

**Affiliations:** 1School of Biosciences, Faculty of Medical and Health Sciences, Taylor’s University, Subang Jaya 47500, Malaysia; 2Faculty of Data Science and Information Technology, INTI International University, Nilai 71800, Malaysia; 3Biofunctional Molecule Exploratory (BMEX) Research Group, School of Pharmacy, Monash University Malaysia, Bandar Sunway 47500, Malaysia; 4College of Pharmaceutical Sciences, Zhejiang University, 866 Yuhangtang Road, Hangzhou 310058, China; 5Centre for Drug Discovery and Molecular Pharmacology, Faculty of Medical and Health Sciences, Taylor’s University, Subang Jaya 47500, Malaysia; 6PAP Rashidah Sa’adatul Bolkiah Institute of Health Sciences, Universiti Brunei Darussalam, Gadong BE1410, Brunei

**Keywords:** atherosclerosis, immune cell, senescence

## Abstract

Atherosclerosis is one of the main underlying causes of cardiovascular diseases (CVD). It is associated with chronic inflammation and intimal thickening as well as the involvement of multiple cell types including immune cells. The engagement of innate or adaptive immune response has either athero-protective or atherogenic properties in exacerbating or alleviating atherosclerosis. In atherosclerosis, the mechanism of action of immune cells, particularly monocytes, macrophages, dendritic cells, and B- and T-lymphocytes have been discussed. Immuno-senescence is associated with aging, viral infections, genetic predispositions, and hyperlipidemia, which contribute to atherosclerosis. Immune senescent cells secrete SASP that delays or accelerates atherosclerosis plaque growth and associated pathologies such as aneurysms and coronary artery disease. Senescent cells undergo cell cycle arrest, morphological changes, and phenotypic changes in terms of their abundances and secretome profile including cytokines, chemokines, matrix metalloproteases (MMPs) and Toll-like receptors (TLRs) expressions. The senescence markers are used in therapeutics and currently, senolytics represent one of the emerging treatments where specific targets and clearance of senescent cells are being considered as therapy targets for the prevention or treatment of atherosclerosis.

## 1. Introduction

Atherosclerosis is associated with the deposition of atherosclerotic plaque composed of accumulated cholesterol, lipids, or other components encapsulated by a fibrous cap on artery walls, leading to pathological intimal thickening with partial or completed blockage of blood flow. It is a typical aetiology of cardiovascular disease (CVD) ranging from mild angina pectoris to fatal myocardial infarction or heart failure. As a result of atherosclerosis, CVD prevails as the leading cause of death worldwide where about 17.9 million deaths reported in 2019 due to cardiovascular-related diseases, including stroke and myocardial infarction [1]. One of the predisposing factors of atherosclerosis is biological aging, in which inflammatory diseases have been frequently manifested in senior citizens. This is due to basal up-regulation of inflammatory mediators, which is closely related to immune cell aging, or known as immunosenescent. Immunosenescence has been highlighted in the progression of atherosclerosis as the function of the immune system runs downhill in terms of impairment of cytokine production, impaired cell-to-cell communication, epigenetic modification [2], decrement of immune cell repertoire, cell repertoire shift, and other factors [3]. 

## 2. Pathophysiology of Atherosclerosis

Typically, the building blocks of atherosclerotic plaque are lipids, particularly low-density lipoprotein (LDLs), and leukocytes, especially monocytes. Initiation of atherosclerosis begins with aberrant endothelial cells coupled with augmented expression of surface adhesion molecules such as intercellular adhesion molecule 1 (ICAM-1), vascular cell adhesion molecule 1 (VCAM-1), and E-selectin. Concomitantly, it was enhanced by proinflammatory cytokines and LDLs which promote monocytes infiltration into the tunica intima layer of blood vessels. LDLs are susceptible to oxidation due to enzymatic modification by lipoxygenases and myeloperoxidase or reactive oxygen species (ROS). The oxidized low-density lipoproteins (oxLDLs) are a damage-associated molecular pattern (DAMP) which are recognized by immune cells and activate the NFκB pathway to signal pro-inflammatory cytokines release including IL-1, TNF-α, and adhesion molecules (VCAM-1, E-selectin and ICAM-I) [4]. In addition, low shear stress contributed to turbulent blood flow as senescent endothelial cells diminish nitric oxide (NO) production, hence it fails to vasodilate the blood vessels [5]. With all stated conditions, endothelial cells and smooth muscle cells are activated and release monocyte-colony stimulating factors (M-CSF) that transform monocytes into macrophages. Macrophages engulf the oxLDL and transform into foam cells. The advancement of inflammation is reinforced with proinflammatory cytokines secreted by polymorphonuclear cells (PMN) including tumor necrosis alpha (TNF-α), interleukin-6 (IL-6), interleukin-8 (IL-8) and interferon (IFN)-γ [6]. Platelet-derived growth factor (PDGF) acts as a mitogen for vascular smooth muscle cells (VSMCs) to proliferate and collagen production to build fibrous cap and encompass lipid core during the initial stage. In later stages of lesion progression, pre-existing inflammation is further amplified by the aging of VSMCs with a necrotic core composed of cell debris and lipids enveloped within the fibrous cap. Eventually, VSMCs undergo apoptosis with significant fibrous cap thinning due to the degradation of collagen and extracellular matrix [7]. As a result of that, the atherosclerotic plaques lose their stability and rupture, causing platelet aggregation or even advancing to detrimental complications such as thrombosis [8].

## 3. The Immune System in Atherosclerosis

Monocytes and macrophages are vital in the formation of atherosclerotic plaque. Both TNF-α and IFN-γ are potent proinflammatory cytokines secreted by macrophages, natural killer (NK) cells, and T-cells. IL-12 and IL-18 have distinct functions in promoting IFN-γ gene transcription in primary human CD4^+^ T cells. It was indicated that both AP-1 and STAT4 activations are needed for IL-12-dependent IFN-γ promoter activation, while IL-18 enhanced the AP-1 binding activity. Surprisingly, the synergy between IL-12 and IL-18 greatly enriched the IFN-γ production of CD4^+^ T cells. The combined effect of both cytokines enhanced the AP-1 binding activity by 20-fold, but not by IL-12 administration alone [9,10,11]. In addition, IFN-γ also triggers macrophages and monocyte to secrete ROS thus leading to cellular oxidative stress. Simultaneously, monocyte chemoattractant protein-1 (MCP-1) is also secreted to attract other macrophages and metalloproteinases to disintegrate the fibrous cap of atherosclerotic plaque [12]. Neopterin [13] and IL-12 [14] are highly expressed in atherosclerotic plaque, which could be remarkable biomarkers for atherosclerosis and CVD detection. As atherosclerotic plaque progresses, neopterin expression is intensified considerably. A stable coronary artery disease (CAD) patient and a non-CAD patient showed minimal neopterin expression in their normal and non-stenotic coronary arteries. On the other hand, neopterin expression was demonstrated to heightened in both early and advanced atherosclerotic plaques and macrophage foam cells in the coronary arteries of stable and unstable CAD patients [13]. Young apoE-deficient mice injected with IL-12 daily exhibited higher levels of antioxidized LDL antibodies and more rapid atherosclerosis than controls injected with PBS only [15]. Nevertheless, targeting the gene encoding IL-12 or vaccines against IL-12 hinders early lesion development through plaque stabilization but is not shown in late lesions [16,17]. 

Dendritic cells (DCs) are the most potent antigen-presenting cells (APCs) in the cell-mediated immune response through upregulated-surface receptors that act as co-receptors in naïve T-cells activation with respective co-stimulatory molecules including major histocompatibility complex MHC I or II. OxLDLs are turned into foam cells through exophagy by mature DCs, which contribute to atherosclerotic plaque formation. The cells create a seal zone coupled with acidification and lysosomal secretion in the contact zone [18]. MyD88 gene is essential in the maturation of DCs through increased antigen-presenting ability and T-cells activation. Mature DCs induced the production of pro-inflammatory cytokines IFN-α and IFN-β, and chemotactic factors CCR5 and CCL5 are upregulated to exacerbate local inflammation by attracting monocytes and T-cells in atherosclerotic lesions [19]. Furthermore, DCs in aortic walls increase the production of TNF-α, IFN-γ, and collagenases MMP to degrade surrounding collagen and extracellular matrix and destabilize the fibrous cap [20]. 

T-cells consist of Th1, Th2, Th17, helper T cells, cytotoxic T-cells, and regulatory T-cells (T reg). They exert both proatherogenic and anti-atherosclerotic effects. In early lesions, the amount of CD8^+^ T cells is little compared to later lesions. Research showed that CD8^+^ T cells have been shifted into the dominant type in later lesions [21]. The synergistic effect of IL-12 and IL-18 on IFN-γ secretion assists in the recruitment of macrophages and differentiation of naive T cells into cytotoxic cells. Macrophages produce MMPs that degrade extracellular matrix while cytotoxic T-cells are equipped with perforin to induce pore formation coupled with granzyme B to activate caspase which led to apoptosis in VSMCs. The synergistic effect of macrophages with cytotoxic T cells is sufficiently potent to disintegrate the fibrous cap. On contrary, Th2 cells act as modulators by negating the production of IFN-γ through the secretion of anti-inflammatory cytokines such as IL-4, IL-5, and IL-13 [22]. IL-4 and IL-13 suppress monocyte activities by impeding the production of inflammatory cytokines, IL-6, TNF-α, and IL-12 [23] and by rendering the secretion of IL-1R antagonists [24]. Although IL-4 and IL-13 are anti-inflammatory in nature, but they also significantly increase the expression of VCAM-1 on the surface of vascular endothelial cells, causing T-cells to adhere to them [25] to facilitate neovascularization [26]. In endothelial cells, IL-4 and IL-13 induce the synthesis of monocyte chemoattractant protein-1 [27], and both HUVEC and dermal microvascular endothelial cells respond to IL-13 [28]. In addition, the release of TNF-α and IL-4 can induce apoptosis of endothelial cells [29]. The profound cytokines from T reg cells are anti-inflammatory cytokines, IL-10 and TGF-β. IL-10 downregulated the expression of IFN-γ, henceforth refraining from monocyte or macrophage activation, releasing cytokine and the upregulation of costimulatory molecules [30,31]. Whereas TGF-β1 was shown to repress the stimulation of IL-1β and TNF-α induced VCAM-1 expression, thus eliciting an athero-protective effect [32,33]. 

B-cells are complementary to T-cells, having both pro-atherogenic and anti-atherosclerotic properties. There are two major subsets of B-cells, which include B1 cells and B2 cells. B1 cell subtypes are further divided into B1-a and B-1b cells. Both subtypes are considered athero-protective by activation of the humoral response against the presence of oxLDL to produce anti-oxLDL IgM antibodies. B1-derived IgM binds to epitopes on apoptotic cells, enabling them to be cleared more efficiently and reduce the inflammation [34] with the assistance of IL-5 which allows proper expansion and releasing antibodies to prevent foam cell formation [35]. However, derived-1a-derived innate response activator (IRA) releases granulocyte-macrophage colony-stimulating factor (GM-CSF) that acts as mitogen of Ly6Chi monocytes, thus aggravating atherosclerosis by inflammatory response [36,37]. A study has demonstrated that mice deficient in IRA B cells are secured against atherosclerosis [38]. In turn, B2 cells will further differentiate into marginal zone B MZ-B cells and follicular B Fol-B cells. Fol-B cells are the predominant B2 population that produces atherogenic IgG that triggers MHC-II expression, therefore recruiting CD4^+^ T cells as well as upregulation of proinflammatory cytokines TNF-α, IL-1β, and chemokines MCP-1, henceforth exacerbating atherosclerosis [39]. Furthermore, it has been shown that Fol-B cells promote Th1 and secrete proinflammatory cytokines [40,41]. T follicular helper cells activate Fol-B cells, which differentiate into germinal center B (GC-B) cells that create GCs [40,41]. As a result of GC-B cells proliferation and affinity maturation, high-affinity antigen-specific IgG and IgE are produced [42]. It has been documented that Fol-B cells are proatherogenic, mainly by secretion of IgG and activating Th1 cells [40,41]. Moreover, the role of MZ-B cells and regulatory B cells in atherosclerosis remains obscure [39].

In summary, the role of monocytes, macrophages, dendritic cells, T-cells, and B-cells in atherosclerosis is depicted in Figure 1.

## 4. Immunosenescence in Atherosclerosis

Changes in the cellular and antibody-related immune response that occurs along with aging are termed ‘immuno-senescence’ [40,41]. Immunosenescence has been reported to associate with several age-related pathologies [42]. The decline in immune protection leads to vulnerability to infections, autoimmune diseases, and low-grade chronic inflammation, specifically termed inflammaging [43,44]. Inflammaging is associated with impaired autophagy [43,45] and ubiquitination [43,46], overproduction of reactive oxygen species (ROS) and reactive nitrogen species (RNS) [47], and release of senescence-associated secretory phenotype (SASP) factors [43,48]. Atherosclerosis is an aging-associated pathology that may be exacerbated by inflammaging [49]. Ideally immune cells function to eliminate senescent cells as part of a protective mechanism against inflammatory responses. However, senescent immune cells have impaired functions and result in the accumulation and persistence of SASP release, promoting aging and age-related diseases [50,51]. Senescent immune cells have been found in the vasculature wall, characterized by increased secretion of proinflammatory mediators from macrophages, DCs, and foam cells that aggravate the atherosclerotic plaque formation [52].

On the other hand, senescent immune cells may also contribute to reducing plaque growth and necrosis. The decline in senescent naive T- and B-lymphocytes with the reduced T-cell receptor (TCR) [53,54] and B-cell receptor (BCR) [38,54] repertoire weakens the adaptive immune response. Moreover, the decline in response for Toll-like receptors (TLR) stimulation upon ligand (antigen) binding by macrophages/monocytes and DCs impede intracellular signaling in inflammatory response activation [55,56,57]. In addition to that, the decline in phagocytic capacity, antigen-presenting, cytokine production, and chemotaxis ability of macrophages hampers its initiation of clonal T-cells proliferation, therefore attenuates the progression of plaque inflammation [40,41]. Senescent immune cells share features of damaged DNA, impaired gene and mitochondrial function [43,58], cell cycle and growth arrest [59], apoptotic resistance, and production of SASP as inflammatory mediators [43,53].

## 5. Features of Cellular Senescence

### 5.1. Morphological Changes

In vitro culture studies showed that senescent cells are characterized by structural changes such as enlarged sizes [60], irregular shape [61], multinucleated (cells containing more than one nucleus) as well as an increased number of lysosomes [62,63]. Other cellular alterations include, darker nucleolus and nuclear chromatin, accumulation of impaired mitochondria, and enlarged endoplasmic reticulum [61]. The shape changes due to reduced expression of scaffolding protein, caveolin, 1 and Rhk GTPase Rac1 and CDC42 [62,64]. Senescent cells form cytoplasmic bridges towards neighboring cells to aid paracrine signaling [62,65,66,67] via direct intercellular protein transfer. Besides that, some senescent cells even have increased granularity as a result of increased vacuoles and lysosomes [68,69].

Other changes that are observed in senescent cells are the decline in nuclear Lamin B1 protein. Lamin B1 is a component of nuclear lamina that acts as a structural protein which surrounds the nucleus, and functions to provide mechanical support and shape to the nucleus and cell [68]. Loss of Lamin B1 protein was demonstrated in an in vivo study of mouse hepatic cells upon induced senescence via ionizing radiation. The decline of Lamin B1 protein is dependent on p53 and/or pRB signaling cascades that are usually activated under cellular replicative stress. The decline in Lamin B1 triggers the downstream inflammatory pathways causing the release of inflammatory mediators such as IFN-β [68,70].

### 5.2. Cell Surface Markers Expression

#### 5.2.1. Intracellular Adhesion Molecule-1 (ICAM-1)

ICAM-1 or CD54 is a cell surface glycoprotein and a member of the immunoglobulin superfamily [71]. It acts as a costimulatory molecule on APCs to activate T-cells by binding to its receptor, lymphocyte function-associated antigen-1(LFA-1). The expression of ICAM-1 or CD54 by the damaged or inflamed vascular endothelial cells is observed to be increased due to elevated stimulation by the pro-inflammatory cytokines released at the atherosclerotic plaque developing site. The ICAM-1 binds to its ligands which include leukocyte function antigen-1 (LFA-1) to initiate the downstream pro-inflammatory signaling pathways [72,73].

Studies showed that ICAM-1 is overexpressed in senescent human fibroblast and human vascular smooth muscle (HVT-SM1) cells. Besides that, immunocytochemical analysis revealed a high expression of ICAM-1 in atheromatous lesions [73,74]. Staining of semi-serial sections of carotid atherosclerotic plaque showed increased p53 expression within the nuclei of foam cells, especially at the ICAM-1 rich areas. Moreover, ICAM-1 is observed to be induced by the elevated expression of p53 in senescent cells which in turn promotes atherogenesis [73,75]. Co-expression of p53 and ICAM-1 is observed in atherosclerotic lesions. It is concluded that p53 activation triggers overexpression of ICAM-1 in senescent cells at atheromatous plaque sites [73]. Several studies reported that CVD patients with diabetes and hypertension who possess high serum levels of ICAM-1 are associated with atherosclerosis and coronary artery calcified plaque (CAC) progression. However, no association was found between serum soluble ICAM-1 and carotid-intima-media thickness (IMT). Hence, high ICAM-1 serum concentration is an early sign of endothelial dysfunction and atherosclerotic plaque [76].

#### 5.2.2. CD36

CD36 is a scavenger receptor found in phagocytes such as macrophages and DCs. It can bind to various ligands including oxidized phospholipids, thrombospondin-1, long-chain fatty acids as well as oxLDL [71]. CD36 initiates oxLDL internalization and foam cell formation which results in loss of macrophage cell polarity and mobility leading to the development of atherosclerotic lesions, promoting atherogenesis [77].

The CD36-mediated internalization of oxLDLs activates phosphorylation of Src family non-receptor tyrosine kinase, specifically lyn, that in turn activates the mitogen-activated protein kinase (MAPK) cascade, specifically Jun-kinase 1 and 2 (JNK). As a result of that, the NF-kB pathway is activated and pro-inflammatory cytokines and chemokines known as SASPs are released [77,78,79,80]. The elevated transformation of lipid laden macrophages into foam cells and persistent SASPs release contributes to atherosclerosis.

A study found a decline in the CD36 expression on aged cardiac M2 macrophages that is normally responsible for plaque regression in the senescence accelerated-prone 8 (SAMP8) mice. It correlates with enhanced M1 and decreased M2 macrophage phenotype polarization that occurs in aging cardiac tissue [81]. This leads to an increased release of pro-inflammatory cytokines (SASP) than anti-inflammatory cytokines, thereby promoting CVD. The macrophages CD36 knockout in ApoE^−/−^ mice study demonstrated decreased aortic atherosclerotic lesion size because of impeded oxLDL uptake. However, the differences in the reduction of lesion size between aged and young mice were not significant [82]. Considering that aging is a risk factor for atherosclerosis, the association of aging, senescent macrophages, and CD36 expression warrants further research. A study presented that CD36 is overexpressed in senescent non-immune cells such as human fibroblast cells that are associated with replicative senescence [83]. However, the exact mechanism of senescent immune cells overexpressing CD36 is unknown.

#### 5.2.3. Urokinase-Type Plasminogen Activator Receptor (uPAR)

Urokinase receptor regulates intracellular signaling pathways in response to extracellular components [84]. It initiates proteolytic activity and degradation of ECM by converting plasminogen into active plasmin leading to the release of fibrin degradation products within the bloodstream [85]. The expression of uPAR is closely linked to senescence [71]. In the atherosclerotic plaque site, the uPA/uPAR/PAI-1 axis is involved in enhanced monocyte recruitment as well as macrophages and foam cell formation [86]. Moreover, soluble uPAR (suPAR), a product of uPAR cleavage upon binding with its ligand is observed to be overexpressed in senescent cells [84,87].

Immune cells such as monocytes, macrophages and activated T-cells are found to express uPAR. uPAR is found to be accumulated in atherosclerotic plaque lesions, causing plaque instability and rupture. High plasma and plaque suPAR level is observed in aged patients especially those with CVD risk factors such as diabetes and elevated blood LDL [88]. High suPAR level is found to correlate with plaque macrophages and monocytes level [88,89].

#### 5.2.4. Beta-2 Microglobulin (B2MG or B2M)

B2MG or B2M is a non-glycosylated surface protein expressed in most nucleated cells. B2M is associated with carotid vascular intimal thickness, and acts as a biomarker of oxidative stress [90] and senescence.

B2M is highly expressed on the surface of lymphocytes and monocytes during chronic inflammation [91]. This aids in upregulating antigen presentation to CD8^+^ T cells, thereby enhancing inflammation and excessive ROS production which promotes atherogenesis [90]. A cytotoxic nanoparticle containing B2MG antibody designed to recognize the senescent cells expressing B2M, has been shown to trigger the release of granzyme B for apoptosis induction [92].

#### 5.2.5. Dipeptidyl-Peptidase 4 (DPP4 or CD26)

DPP4 is a cell surface glycoprotein expressed on senescent cells. It is involved in cleaving various peptide substrates such cytokines and growth factors [71]. Immune cells that express DPP4 include T-cells, B-cells, DCs, and macrophages [93]. DPP4-expressing lymphocytes and monocytes were found predominantly in aged individuals [94] as well as in mouse models fed with a high-fat diet [95,96]. DPP4 expression on DCs and macrophages is shown to promote T-lymphocytes activation and proliferation via direct binding to the T-cells with adenosine deaminase (ADA) protein [93]. Overexpression of DPP4 was found to correlate with upregulation of other senescence features such as elevated p16 and p21 signaling, SA-βgal activity and ROS in pre-senescent fibroblast cells, WI-38 [94]. The DPP4 inhibition was found to correlate with decline in TNF-α, IL-6, and MCP-1 as well as monocyte activation and chemotaxis, denoting its role in inflammatory responses [95,96]. A DPP-4 inhibitor, alogliptin, was found to inhibit MMP expression that slows tissue degradation and remodeling at atherosclerotic sites [97,98]. Besides that, the DPP-4 inhibitor is also found to inhibit macrophages infiltration and accumulation at inflamed vasculature sites [93]. The atherogenic role of DPP4 in CVDs is further supported by a diabetic ApoE^−/−^ mouse model study which showed reduced atherosclerotic lesion size upon administration with alogliptin [97].

### 5.3. Cellular Metabolism and Secretory Phenotype

#### 5.3.1. Increased Metabolic Activity

Senescent cells are shown to exhibit high metabolic activity. In a pre-senescent endothelial cell, lactate dehydrogenase is upregulated, but pyruvate dehydrogenase and malate-aspartate shuttle are downregulated [99]. This leads to the diversion of pyruvate from the electron transport chain (ETC) and aerobic respiration. It causes vascular endothelial cells to produce adenosine triphosphate (ATP) anaerobically. These metabolic shift causes a decline in NAD^+^/NADH ratio that inhibits the enzyme, sirtuin (SIRT) and increases AMP/ATP and ADP/ATP ratio [99] that triggers the adenosine monophosphate-activated protein kinase (AMPK) pathway which in turn activates the phosphorylation of p53. Activation of p53 causes cessation of cell replication and growth, thereby promoting cellular senescence [100].

Aged CD8^+^ CD28^−^ T-cells have decreased sirtuin-1 (SIRT1) which activates the p53 signaling pathway leading to cell-cycle arrest [99]. SIRT1 is involved in establishing a homeostatic balance in glucose and fat metabolism. It is shown to protect cardiomyocytes and endothelial cells from oxidative stress as well as prevents VSMC hypertrophy, therefore alleviating atherosclerotic plaque progression [101]. Senescent T-lymphocytes are demonstrated with reduced SIRT-1 specifically in the CD28^−^CD8^+^ T-cells contributing to chronic obstructive pulmonary disease (COPD) [102]. Besides its metabolic role, SIRT-1 decline is shown to cause a decrease in Forkhead box transcription factor 1 (FOXO1) expression and function that enhanced the pro-inflammatory cytokines’ release from CD28^−^CD8^+^ T-cells [103]. The FOXO1 functions to control macrophages oxLDL uptake and causes arterial wall calcification, thereby promoting atherogenesis [104]. Therefore, SIRT-1 decline in senescent T-cells could slow the atherosclerosis plaque progression.

#### 5.3.2. Senescence-Associated Secretory Phenotype (SASP)

Senescent cells release SASP factors which comprise inflammatory cytokines (interleukins, IL-1, IL-1β, IL-6), chemokines (growth-regulated alpha protein 1 (GRO1)), proteases such as MMPs, vascular endothelial growth factor (VEGF), signaling molecules such as NO and hormones such as prostaglandin E2 [50] that induces proinflammatory responses [50,67].

Senescent CD14+CD16+ monocytes at the atherosclerotic plaque site were found to secrete significant levels of VCAM-1 for adhesion and diapedesis of monocytes and lymphocytes into the inflamed site [105]. Meanwhile, senescent M1 macrophages at the atherosclerotic plaque site have been shown to release SASPs which enhances the oxidative stress in plaque, leading to its instability and rupture. An increased number of senescent Th1 and Th17 CD4^+^ helper T-lymphocytes were found at the carotid atherosclerotic plaque site secreting high amounts of IL-17, IL-21, IL-23 that accelerate the plaque progression [49].

### 5.4. Lysosomal Activity

#### 5.4.1. Lysosomal Alteration

Senescent cells are observed to have increased size and number of lysosomes [106,107]. As the abundance of lysosomes increases, the accumulation of oxidized lipid-laden lipofuscin in lysosomes also increases. An added burden is the mitochondrial damage that overly releases ROS causing more lipofuscin accumulation in lysosomes during stress-induced senescence [106].

The mitochondrial structural changes are stated to be associated with reduced lysosomal functionality particularly, due to the accumulation of lipofuscins. This association is known as the lysosomal-mitochondrial axis which also plays a role in cellular senescence. Lipofuscin is an aggregate of highly oxidized lipids [108], made of cross-linked protein residues inside the lysosome, derived from iron-catalyzed oxidation of lipids and they are non-degradable by lysosomal hydrolytic enzymes [109]. It disrupts the autophagy-lysosome pathway, therefore damaged organelles fail to be cleared off and instead leads to persistent inflammation [110]. The lipofuscin-loaded lysosome acts as a sink for the newly synthesized lysosomal enzymes, resulting in a reduced lysosomal activity. The decline in lysosomal activity in turn promotes enhanced oxidation of lipids and protein, thereby giving rise to more lipofuscin production [106,111]. Lipofuscin is observed to be accumulated in lysosomes of aged cardiomyocytes and ischemic cardiomyopathy [112]. Replicative senescent T-cells were shown to have accumulated lipofuscin which increases with aging [113].

Macrophage foam cells are known to have increased lysosomal pH levels. Reduction in lysosomal acidification is not optimum for the hydrolytic enzymes and would further reduce their autophagy capacity. Therefore, macrophages with damaged organelles are not repaired but subsequently enter a senescence state. Therefore, the capacity of macrophage foam cells in antigen presentation and phagocytic capacity are greatly reduced during senescence. Hence, it leads to reduced responsiveness in inflammation during senescence [106,114]. A study revealed that senescence-induced macrophages possess increased lysosomal pH and are associated with aging [115]. Moreover, lysosome impairment causes inefficient hydrolysis of lipoprotein and oxLDLs in the macrophage foam cells leading to defective reverse cholesterol transport promoting atherosclerotic plaque progression. Hence, individuals with lysosomal storage disease (LSD) have a higher risk of developing atherosclerosis at an earlier stage [116].

#### 5.4.2. SA β-Galactosidase (SA-βgal) Activity

SA-βgal is a hydrolase enzyme that actively hydrolyses β-galactosidase into monosaccharides in senescent cells only at a pH of 6.0 [59,117]. It maintains cellular acidity to facilitate damaged cell degradation. SA-βgal activity is found abundantly in pre-senescent cells due to its increased lysosomal content [62,118,119]. The high SA-βgal in senescent cells denotes high lysosomal β-galactosidase in the cell to meet its high glycolytic demand. A recent study has shown that senescent CD8^+^ T-cells with high SA-βgal activity have a positive correlation with aging [120].

Another study has also reported high SA-βgal activity within the mature macrophages and aged cardiomyocytes [121,122]. A combination of both SA-βgal activity and lipofuscin can be markers [64]. It is noted that high expression of SA-βgal, p15, p16, p21, p53, and ARF is found in endothelial cells and VSMCs of human atherosclerotic plaque [60]. An autopsy study has revealed elevated SA-βgal activity in endothelial cells within the atherosclerotic plaque that has developed within the coronary artery [123]. It is stated that increased SA-βgal in association with increased IL-6, p16, and p21 were detected in VSMC, contributing to the progression of atherosclerosis [123,124].

#### 5.4.3. The Mammalian Target of Rapamycin (mTOR) Activity

The mTOR is a protein kinase that controls cellular transcription and protein synthesis to promote cell proliferation and differentiation. It is also involved in SASP synthesis and regulation [62]. The kinase, mTORC1 is present in lysosomes. It inhibits lysosomal biogenesis by inhibiting transcription factor EB (TFEB)which is part of the melanocyte-inducing transcription factor (MITF) family [125]. The transcription factor (TFEB) regulates gene expression involved in the production and autophagic pathways of lysosomes by binding to the promoter regions of the genes. Hence, inhibition of TFEB by mTORC1 reduces lysosomal biogenesis and proliferation [126]. On the other hand, mTORC1 increases mitochondrial biogenesis which is crucial to compensate for mitochondrial damage and dysfunction that occurs with aging [127]. This affirms that mTOR has a multifaceted role not just in terms of lysosomal and mitochondrial biogenesis but also in age-related pathologies.

When the mTOR is inhibited, it fails to be bound by its accessory protein raptor which is needed for the downstream phosphorylation of S6K1 and 4EBP1. The phosphorylated 4EBP1 regulates the translation of IL-1α [62,128] and MAP kinase–activated protein kinase 2 (MAPKAPK2) [62,129] that inhibits ZFP36L1 causing SASP mRNA degradation. That in turn combats the release of pro-inflammatory mediators. Hence, mTOR indirectly controls SASP secretion in senescent cells. It is stated that mTOR complex 1 (mTORC1) is elevated in senescent cells and associated with autophagy [130]. In general, increased mTORC1 activity is associated with increased SASP release in senescent cells.

The mTOR activation is stated to inhibit autophagy-related genes (Atg). For instance, mTORC1 inhibits Atg1/Ulk1 which generally initiates the autophagic pathway by forming the Atg1 protein kinase regulatory complex for downstream signaling [131]. It also disrupts the Ulk1/AMPK pathway that regulates autophagy by phosphorylating Ulk1 Ser 757 [132]. Hence, mTOR activation inhibits autophagy. Inhibition of autophagy is associated with enhanced atherosclerosis. This is because, with exiting persistent oxidative stress within the atherosclerotic plaque, added by a halted autophagic pathway which acts as a part of the host defense mechanism, cells with a damaged lysosomal membrane (due to high ROS production) end up in apoptosis [133]. That in turn aggravates the inflammation at atherosclerotic lesions.

Elevated mTOR activity is stated to be involved in the early stage of atherosclerotic development as increased macrophage numbers with mTOR activation were observed at the atherosclerotic plaque [134]. The mTOR hyperactivation in macrophage foam cells enhances macrophages and VSMCs migration into tunica intima during atherosclerotic events [135]. Besides that, mTOR signaling activation in vascular endothelial cells (VSMCs) is shown to reduce the klotho protein expression where it promotes local phosphate uptake by VSMC, thereby induces vascular calcification and compromising the endothelial function. Therefore, mTOR inhibitors are also known as klotho activators which could delay the onset of age-related CVD risks [136,137].

The function and activity of mTORC1 in senescent immune cells, more so during aging, is not well studied. However, a study showed that mTOR inhibitor rapamycin can reverse T-lymphocyte senescence by enhancing its autophagic capacity and inhibiting its SASP release. This is supported by another study where rapamycin decreased the serum SASP marker, IL-6 in the older population with coronary artery disease [138,139].

### 5.5. Cell Cycle Arrest and Apoptosis Resistance

Cell cycle arrest is an irreversible halt between the cell cycle checkpoints/phase transitions either in the interphase or mitotic phase to allow DNA repair and homeostasis restoration. Senescent cells have restricted proliferative capacity. Senescent cells are resistant to extrinsic and intrinsic apoptosis due to the upregulation of anti-apoptotic proteins, such as BCL-2 family proteins such as BCL-W and BVL-XL. Therefore, BCL-2 family inhibitors may serve as senolytics which can induce apoptosis of senescent cells. The cell cycle arrest is regulated by several molecular pathways including p16^INK4a^/Rb and p53/p21^CIP1^ [62].

The p21^CIP1^/p53 and p16^INK4a^/Rb pathways are crucial for the induction of growth arrest. p53 is usually bound to mouse double minute 2 homolog (Mdm2) which is an E3 ubiquitin protein ligase that functions in ubiquitination and proteasomal degradation. When p53 is phosphorylated or activated, it is released from Mdm2 where it stimulates the transcription of p21^CIP1^(CDKN1A). p21 inhibits cell cycle regulator, cyclin-dependent kinase (Cdk2) [47,117] and results in activation of retinoblastoma (Rb) and cell cycle arrest [52]. Activated Rb binds to the E2 factor (E2F), thereby blocking its transcriptional activity of DNA replication genes. Interestingly, p53 is shown to increase in advanced human atherosclerotic plaque. In vitro studies have reported that inhibitors of p53 and pRB are shown to suppress cell cycle arrest and senescence of VSMCs at the atherosclerotic plaque site [47,117].

CDKN2A locus also known as INK4A and alternative reading frame (ARF) encodes for two tumor suppressor genes, namely cell cycle inhibitor, p16^INK4A,^ and p53 activator, p14^ARF^. p16^INK4A^ inhibits the cyclin-dependent kinase complexes, CDK4 and CDK6 for long-term cell cycle arrest whereas p14^ARF^ regulates the stability of the p53 gene. Normally genes from CDKN2A loci are expressed at low levels in young tissue but are found to be depressed in aged tissues but its exact mechanism is unknown [62]. Gene polymorphism studies revealed that low expression of p16^INK4A^ and p14^Arf^ (p19^ARF^ in mice model) has a positive correlation with increased atherogenesis [140]. However, whether senescent cells downregulate or upregulate its expression level leading to either accelerated or inhibited atherogenesis is unclear.

p16I^NK4a^ is observed to have been expressed in atherosclerotic plaque, in CD68-positive macrophages, and VSMCs [141,142]. p16^INK4a^ deficiency in macrophages is stated to have declined in its release of inflammatory factors via inhibition of STAT1 and NF-kB signaling pathways [141]. Overexpression of p16^INK4a^ in macrophages is shown to lead to pathologies associated with thrombus occlusion as part of atherosclerotic complications. However, whether senescent macrophages are associated with elevated p16^INK4a^ is unclear [141,143] but long-term expression of p16^INK4^a is shown to cause irreversible cell-cycle arrest and promote aging [144].

## 6. Regulation of Immune Cells Senescence in Atherosclerosis

Several immune cell types, including monocytes, macrophages, DCs, and T- and B-lymphocytes are involved in promoting senescence in atherosclerosis development

### 6.1. Monocytes, Macrophages, and Foam Cells

All three subsets of monocytes, classical (CD14++CD16−), intermediate (CD14++CD16+) and non-classical (CD14+CD16++) are observed to exhibit hallmarks of senescence [145,146]. However, one of the three subsets of monocytes, non-classical (CD14+CD16++) is observed to be elevated and highly secrete IL-8 and TNF-α in the blood plasma of the elderly population [147,148]. Non-classical (CD14+CD16++) monocytes are stated to highly express membrane-bound IL-1α and have upregulated NF-kB signaling, which is reminiscent of the senescence feature, SASP. The accumulation of non-classical subset and CD16^+^ monocytes in the elderly population is reported to cause atherosclerosis due to their SASP secretion. The non-classical subset is reported to have the least proliferative capacity than the other two subsets, shorter telomere length, and anti-apoptotic properties besides a high concentration of cellular and mitochondrial ROS. Besides TNF-α and IL-8 [149], SASPs secreted from non-classical monocytes include CCL3, CCL4, CCL5, IL-6, IL-1β [147,148]. It is the subset that expresses the highest level of membrane-bound IL-1α and causes upregulation of NF-kB activation which is then followed by intermediate and classical subsets [150].

Several studies stated, most CVD patients are detected with a high subpopulation of non-classical monocytes, CD14+CD16+. CD14+CD16+ monocytes are demonstrated to have upregulated phagocytic activity as well as TNF-α and IL-1β secretion [145,146]. A study revealed senescent CD14+CD16+ monocytes highly express proatherogenic chemokine receptors, CCR2, CCR5, CCR7, and CX3CR1 as well as endothelial adhesion molecules, VCAM-1 and ICAM-1 [149]. These particular subsets of monocytes are also stated to have increased endothelial adhesion capacity [105]. The deficiency of the mentioned subset of monocytes is shown to be associated with increased fibrous cap thickness [21,105]. Moreover, senescent monocytes highly secrete inflammatory chemokines and chemokine receptors by such as CCL-2, CCL-3, CCL-4 and CCR-2, CCR-5, CCR-7, and CX3CR-1 that act as chemo-attractants, contributing to atherosclerotic plaque development [52]. The intermediate (CD14++CD16+) subset of monocytes is shown to upregulate the expression of CD74, a surface marker that acts as a receptor for macrophage inhibitory factor (MIF) [151]. It enhances the MIF expression in macrophages [152] which in turn promotes chemokine, and CCL-2 secretion from the endothelial cells. It serves to upregulate monocyte-endothelial adhesion and monocyte arrest and chemotaxis [153].

Senescent monocytes also release SASPs to recruit more peripheral monocytes for chemotaxis, diapedesis, and transmigration via the endothelium into inflamed vascular sites [154]. This is enhanced by senescent vascular endothelial cells (VECs) that decline in their NO secretion leading to compromised vascular integrity and protective functionality. Moreover, senescent VECs increase their secretion of monocyte chemokine protein-1 (MCP-1) that in turn recruit more peripheral monocytes to the endothelium. This leads to the formation of foam cells at an elevated rate, accelerating atherosclerotic plaque progression which potentially causes acute coronary disease such as myocardial infarction (MI) [155]. In addition to that, Klotho protein involves in regulating oxidative stress by reducing the expression of TNF-α from monocytes/macrophages. It has been reported that in patients with atherosclerosis, Klotho protein expression was observed to be reduced [156]. However, its expression level from senescent monocytes/macrophages is unclear. 

Macrophages are the most abundant immune cells within atherosclerotic plaque hence it is the major contributor to plaque formation and progression in terms of cholesterol efflux, inflammation, necrosis initiation, and ECM degradation. Aged macrophages were found to have low expression of TLRs for antigen presentation to the effector T-cells to initiate inflammatory responses [157,158]. In an ApoE^−/−^ mice model study, macrophages TLR 9 deficiencies have reduced atherosclerotic lesion size and vascular inflammation due to low accumulation of macrophages and expression of VCAM-1, ICAM-1, TNF-α and MCP-1 at the atherosclerotic plaque site [159,160].

A study has supported that senescent macrophages have impaired cholesterol efflux due to the downregulation of *ABCA1* and *ABCG1* genes as well as enhanced macrophage polarization into pro-angiogenic and disease-promoting phenotype which promotes inflammation, thereby accelerating the formation of foam cells and plaque progression [158,161]. In addition, senescent macrophages are able to polarize into M1 phenotypes (which include highly expressed SASP factors such as TNF-α, IL-6, IL-1β, CCL2, and collagenase factor, MMP 9) that accelerate the atherosclerotic plaque progression [161].

There are three subtypes of macrophages present in atherosclerotic plaque: resident-like macrophages, inflammatory macrophages, and TREM^hi^ macrophages [162]. Cardiac resident-like macrophages achieve a steady state as one ages, thereby it is compensated by recruited macrophages. However, resident-like macrophages in atherosclerotic plaque are shown to express several senescence markers including CXCL4, CCR2, and CXC3R1 [163]. Inflammatory macrophages are able to secrete SASPs factors including IL-6, TNF, and chemokines (CXCL2, CCL2, CCL3, CCL4, CCL5, CXCL10) [162]. It also highly expresses costimulatory ligands, CD80 that enhance antigen presentation and T-cells recruitment and inflammatory secretion at the atherosclerotic plaque [164]. Foamy TREM^hi^ macrophages have high expression of surface glycoprotein, CD9 that activates (PI3K)/Akt/mTOR signaling pathway associated senescence which aggravates atherosclerotic plaque inflammation [164,165]. In addition, these foamy TREM^hi^ macrophages also highly expressed fatty-acid scavenger receptors (CD36), that accelerate foam cell formation and atherosclerotic plaque expansion in size [77,162]. M2 phenotype macrophages are observed within atherosclerotic plaque and are stated to enhance the plaque rupture due to MMP-9 release that degrades type IV collagen at the fibrous cap, leading to plaque rupture [166].

Interestingly, MicroRNA-33 (miR-33) is frequently found in aged macrophages. MiR-33 is an inhibitor for the cholesterol transporter, ABCA1 [161,167]. Several in vivo studies have proved that mice with knockout of the *miR-33* gene had high plasma HDL levels, reduced plaque size, and decreased inflammatory response [167,168,169,170]. Similarly, inhibition of miR-33 in aged macrophage culture has alleviated vascular endothelial cell proliferation [161]. In one pathogenic study, the upregulation of miR-33 was triggered by *Mycobacterium tuberculosis.* That led to impairment of cholesterol efflux because accumulated cholesterol within the infected macrophages serves to be the source of nutrient for the bacterium [171]. It can therefore be said that patients with comorbidity, especially with TB infection are at high risk for atherosclerosis. However, the exact role of miR-33 expression in senescent macrophages remains unclear.

Other interesting miRNA associated with atherosclerosis is miRNA-126a. It is expressed by senescent macrophage, which is also known as p16 (Ink4a)- and β-galactosidase positive (p16^+^/β-gal^+^) subset [172]. miR-126a is known to promote telomerase activation in senescent macrophages besides promoting increased SA-β-gal activity and expression of cell cycle inhibiting genes, p53 and p16. Upregulated expression of miR-126a can trigger SMAD family member 3 (Smad3), thereafter the downstream NF-kB pathway (Smad3/NF-κB signaling) will activate the telomerase and promote M2 to M1 macrophage polarization within atherosclerotic plaque. As stated earlier, the pro-inflammatory M1 phenotype macrophages promote atherosclerotic plaque development via its secretion of inflammatory mediators that in turn cause vascular endothelial inflammation [173].

The involvement of senescent macrophage in atherosclerosis is affirmed by its role in secreting MMPs to promote plaque instability, elastic fiber degradation and fibrous cap thinning, therefore, accelerate the lesion rupture and thrombosis [52,140,155]. The high expression of MMPs from senescent macrophages and foam cells cause ECM degradation at the atherosclerotic plaque site which leads to weakening of the vascular wall [140]. Moreover, heathy macrophages perform efferocytosis to engulf and clear the cell debris from apoptotic cells to reduce the atherosclerotic plaque burden [174]. However, senescent alveolar macrophages are reported to have reduced efferocytotic capacity that promote sustained inflammation at the atherosclerotic plaque site [175]. 

### 6.2. Dendritic Cells (DCs)

There is scarcity in evidence of DCs senescence associated with SASP release and its role in atherosclerosis. Nevertheless, aging is associated with a reduced capacity of DCs to bind with T-cells [176]. Age-related changes which cause dysregulation of the immune responses are believed to be one of the driving factors of chronic inflammation commonly associated with the elderly population [176,177]. These changes include a low number of peripheral DCs, reduced capacity for chemotaxis and phagocytosis as well as antigen presentation to the effector T-cells [178]. Studies have reported chronic inflammation that is paired with aging during the atherogenic response to be associated with the occurrence of CVD [179,180].

Elevated merocytic DCs subsets (CD8α^−^CD11b^−^) are shown to be correlated with aging where they were shown to impair T-cell priming capacity due to low expression of MHC class 1 surface molecules [181]. Another two subsets of DCs; CD11c^+^ CD11b^−^ and CD11c^+^ CD11b^+^ are observed to be accumulated within the atherosclerotic plaque [182,183]. These CD11c^+^ CD11b^+^ subset of DCs shown to have highly expressed of pro-inflammatory and pro-atherogenic mediators including CCL-2, IL-6, IL-1β. Besides that, high expression of CD36, TLR 2 and TLR4, IL-12, and IL-6 are associated with atherosclerotic plaque formation [184].

The population of CD11c^+^ DC is seen to increase with age in a murine model study [185]. Studies showed that antigen presentation and T-cells priming declined in immuno-senescent DCs [186] and that could inhibit the atherosclerotic plaque progression. CD11c^+^ DC which is predominantly found in atherosclerotic plaque secretes VCAM-1, CCL-2, and scavenger receptors, LOX-1, CD36, and CD205.

### 6.3. T-Lymphocytes

Senescent T-cells have been shown to have activated p38 mitogen-activated protein kinase (p38-MAPK) signaling pathway and expressed IL-18, IFN-γ, and CCR-7 that in turn increased the abundance of CD4^+^/CD8^+^ (TEMRA) cells. Many senescent TEMRA cells are found to be present within the atherosclerotic plaque [52,157,187,188]. The overactivation of the p38-MAPK signaling pathway triggers TCR signaling and IFN-γ expressions of T-cells that aggravates atherosclerotic plaque development [189].

Meanwhile, accumulation of differentiated CD8^+^CD28^−^ T-cell is observed in the elderly population with coronary atherosclerotic plaque [190]. The low expression of CD28 leads to defective T-cells function and responsiveness towards oxLDL antigen presentation by APCs. Moreover, senescent CD4^+^CD28^−^ T-cells are reported to be associated with IFN-γ secretion and this subset is found elevated in patients with unstable angina [190]. Senescent CD4^+^CD28^−^ T-cells are also observed to highly secrete CCR5, CCR7 and CXCR1 that promotes inflammation at the atherosclerotic plaque site [191]. 

### 6.4. B-Lymphocytes

Studies have shown that the late memory B-cell (IgD^−^/CD27^−^) subset expresses the highest level of SASP biomarkers which includes TNF-α, IL-6, and IL-8 and pro-inflammatory miRNAs (miR-155, miR-16, and miR-93) [192]. MiR-155 promotes monocyte differentiation, and accelerates foam cell formation besides down-regulating NO synthase expression which leads to endothelial inflammation [193]. However, it is also capable to reduce oxLDL uptake by down-regulating CD36 and LOX-1 expression [194]. MiR-155 is more commonly detected in patients experiencing acute myocardial infarction and unstable angina than in patients experiencing ordinary chest pain syndrome. High miR-16 is detected in patients experiencing peripheral artery disease [195,196]. In contrast, there is research indicating that elevated miR-16 alleviates atherosclerotic progression as it inhibits pro-inflammatory cytokines, IL-6, and TNF-α release but enhances anti-inflammatory cytokines, IL-10 release from foam cells [196]. MiR-93 is reported to inhibit ABCA1 expression, hence it leads to impaired macrophage cholesterol efflux, promoting atherosclerosis progression [197].

## 7. Therapeutic Strategies Targeting Immune Senescent Cells in Atherosclerosis

Targeting immune senescent cells for apoptosis (senolytics) and inhibition of SASPs secretion (senomorphics) could serve as potential as one of the potential therapeutic strategies in the prevention and treatment of atherosclerosis.

Anakinra is a recombinant IL-1 receptor antagonist that blocks IL-1β binding with its receptor, IL-1R on T-cells [161]. Studies showed that IL-1Ra (IL-1R inhibitor) inhibits the development of atherosclerosis lesions [161]. Phase two trial demonstrated Anakinra to successfully reduce C-reactive protein (CRP) and IL-6 [198].

Meanwhile, Canakinumab and Gevokizumab are monoclonal antibodies that selectively bind to IL-1β leading to the formation of a complex which prevents the IL-1β from binding to its receptor, IL-1R1, thereby blocking the IL-1 signaling cascade [161]. Canakinumab has shown promising effects in alleviating atherosclerosis during the phase 3 clinical trial. [199].

Chimeric antigen receptor (CAR) T cells target uPAR expressed on senescent monocytes and macrophages [200]. The binding of CAR T cell with uPAR on senescent immune cells triggers the intracellular inflammatory signaling cascade leading to effector T-cells activation and proliferation, eventually leading to senescent cell death. There are several interventions in administering the genetically modified T cells that express chimeric antigen receptors (CAR) that consist of antigen binding domains specific towards oxLDL. Patients with high risks for atherosclerosis such as those with a family history of MI were revealed to have reversed and/or prevented atherosclerosis or CVD as their Treg CD4^+^ T-cells were obtained and transduced with lentivirus for CAR expression before reintroduced into their circulation. This treatment is said to have a long-term benefit as memory Treg cells against the oxLDL and uPAR can be developed [201].

Several B2MG inhibitors including suramin and doxycycline are shown to bind specifically to the B2MG via molecular docking studies. Hence, it is suggested to be a novel therapeutic antagonist in atherosclerosis treatment as it reduces the concentration of human B2M protein availability for the aid of inflammation [202].

Meanwhile, senescent macrophages have increased expression of *miR-33* that led to downregulated *ABCA1* and *ABCG1* genes which in turn caused impaired cholesterol efflux. Moreover, silencing or inhibition of *miR-33* could prevent its suppression of *ABCA1* and *ABCG1* which may in turn inhibit atherosclerosis progression [148,161].

## 8. Conclusions and Future Directions

Many inflammatory cytokines and chemokines are involved in atherosclerosis where targeting one specific mediator might not be effective as the rest of the cytokines level might not be targeted and its effects in triggering the inflammatory response may be significant. Therefore, finding the specific antigens/epitope that triggers the immune response could be targeted for antibody production. For instance, antibodies targeting the LDL antigens or senescence immune cells (macrophages, foam cells, T-cells, B-cells) or its markers released from senescent immune cells could serve as potential strategies for the control of senescence in atherosclerosis. There is less investigation on immune cell senescence particularly using human models whereas mice models are mostly being studied for atherosclerosis. However, it is required of specific imaging technologies that are yet to exist in detecting arterial plaque/atheroma and to directly track the senescent cell population from the plaque site rather than isolating it from peripheral blood collection. This helps to evaluate the contribution of immunosenescence to cardiovascular disease risk.

## Figures and Tables

**Figure 1 ijms-23-13059-f001:**
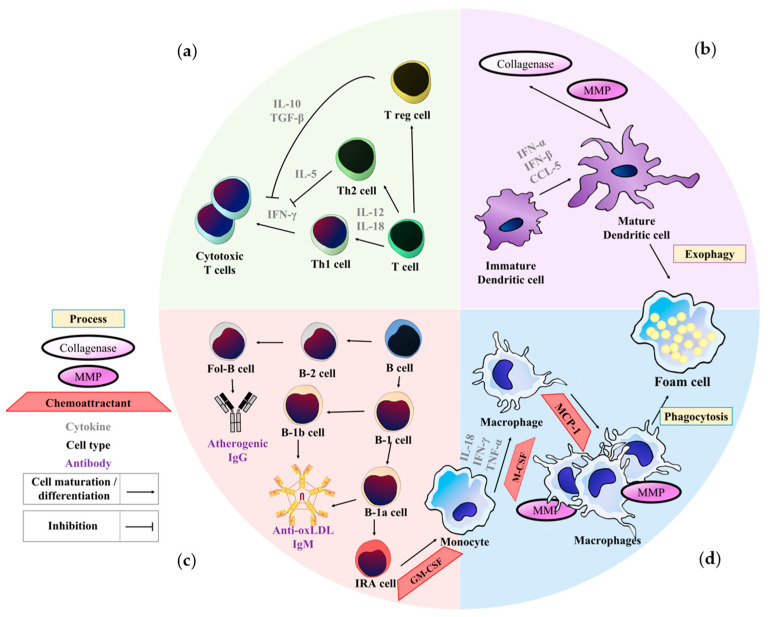
Role of monocytes, macrophages, dendritic cells, T-cells, and B-cells in atherosclerosis: (**a**) T-cells in atherosclerosis: Existence of IL-12 and IL-18 trigger Th1 skewing and secretion of IFN-γ to recruit cytotoxic T cells [9,10,11]. IL-5 secreted by Th2 cells [22] with IL-10 and TGF-β secreted by regulatory T cells (T reg) inhibit the function of IFN-γ [30,31]; (**b**) Dendritic cells in atherosclerosis: IFN-α, IFN-β and CCL-5 secreted by dendritic cells stimulate maturation of immature dendritic cell into the mature dendritic cell [19]. Mature dendritic cells produce metalloproteinase (MMP) and collagenase to destabilize atherosclerotic plaque. Mature dendritic cells also take up oxLDL by exophagy and transform into foam cells [20]; (**c**) B-cells in atherosclerosis: B-cells differentiated into 2 main subtypes, B1 cells, and B2 cells. In B1 cells, it further differentiates into B-1a and B-1b cells, both secrete athero-protective IgM. B-1a cell derived from B-1a cell secretes GM-CSF to enhance the proliferation of Ly6C monocytes. In B-2 cell, Follicular B cell (Fol-B) produce atherogenic IgG [39]; (**d**) Monocytes and macrophages in atherosclerosis: The presence of potent inflammatory cytokines, IL-18, TNF-α and IFN-γ, they initiate atherosclerosis by activating monocytes and macrophages. Macrophage colony-stimulating factor (M-CSF) monocytes stimulate the differentiation of monocytes into macrophages. Monocytes chemoattractant protein-1 (MCP-1) further attract other macrophages to phagocytose oxLDL to become foam cells as well as metalloproteinases (MMP) to degrade the fibrous cap of atherosclerotic plaque [12].

## Data Availability

Not applicable.

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
