# Peer review of "Targeting Immune Senescence in Atherosclerosis"

_ijms, 2022, doi:10.3390/ijms232113059_

Round 1
Reviewer 1 Report
Lee et al have presented an interesting review of an understudied topic. The authors cover many of the well described players in both atherosclerosis and senescence, and discuss both the reported and potential interactions among the various signaling pathways.
Overall, many sections of the manuscript were hard to understand and require extensive editing for English language. The topic is interesting and the references cited are adequate. Several instances are noted where the authors might be overstating the cited literature, such as on line 76 when they discuss pre-existing inflammation being amplified by vascular muscle cell aging. This is hypothetical and was not shown in the citation, but the authors state it as fact. There are multiple instances of this throughout the manuscript.
Author Response
We would like to thank you for your kind comments and suggestions. We have re-looked and make appropriate amendments by correcting all spelling (in British English) and grammatical errors by using “Track changes” function. We also ensure all references were properly cited according to cited literature.
Reviewer 2 Report
This review on targeting immune senescence in atherosclerosis is an interesting and timely subject as much attention is being directed to the role of inflammation in aging, especially in cardiovascular disease states. This manuscript seeks to inform the health science communities of the processes.
Much effort has gone into the production of the manuscript by the large group of authors. However, the manuscript does not provide a clear overall understanding of the processes involved among many difference cells and factors. It is overly detailed and long. Many unimportant details are included. Sentences are complicated, incomplete and difficult to understand. A thorough rewriting is needed. Use of an editing program is strongly suggested.
A few examples of the problems, among many, are provided:
Lines 92-94. Lines 120-128. Lines 130-141. Lines 163-164. Lines 205-212. Line 285. Line 428 – Cytokines are small proteins; ROS and RNS (undefined) are not cytokines. Line 473 – Why is SA-b-gal activity elevated in healthy subjects?

Author Response
We would like to thank you for your kind comments and suggestions. We have re-looked and make appropriate amendments by summarize important information about immunesenescence in atherosclerosis processes involving difference cells and factors. As suggested, many unimportant details have been removed for better understanding and clarity. Full term/name will be defined in the first time before using abbreviation in subsequent text. Appropriate amendments by correcting all spelling (in British English) and grammatical errors were performed by using “Track changes” function.
Round 2
Reviewer 1 Report
Vellasamy et all have greatly improved their manuscript, “Targeting Immune Senescence in Atherosclerosis.” There are a few minor points that require clarification.
Minor concern:
On line 147 the authors write “Th17 is said to secrete IL-17A, IL-17F and IL-22, but its role in atherosclerosis remains controversial [22].” But on line 480, the authors then write, “Increased number of senescent Th1 and Th17 CD4+ helper T-lymphocytes were found at the carotid atherosclerotic plaque site secreting high amounts of IL-17, IL-21, IL-23 that accelerates the plaque progression [48].” It would seem that these two points are contradictory.
Examples of English language problems:
On line 762, “A study proved that senescent macrophages have impaired…” Studies can only support or refute, but they cannot prove.
There multiple instances of incorrect verb tense used such as on line 111: “Likewise, dendritic cells were the most dominant antigen-presenting cells in cell-mediated immune response by upregulation of cell-surface receptors that act as co-receptors in naïve T cells activation with respective co-stimulatory molecules, major histocompatibility complex MHC I or II.” It should read, “dendritic cells are”, and calling DCs the most dominant is not correct. I would think that the authors mean that DCs are the most potent APCs.
There are inconsistencies in the use of singular and plural nouns such as on line 212: “characterized by increased secretion of proinflammatory mediators from macrophage, dendritic cells, and foam cells that aggravates the atherosclerotic plaque formation [51].” Macrophage should be plural.
There are still numerous instances of awkward subject/object-pronoun disagreement, such as this on line 825: “The involvement of senescent macrophages in atherosclerosis was further supported when it can secrete….”
There are still numerous instances of incorrect English language usage, such as line 907: “Senescent T-cells have shown to activate p38… “ I suggest something like “have been shown to have activated”… which sounds more appropriate.
Author Response
1. On line 147 the authors write “Th17 is said to secrete IL-17A, IL-17F and IL-22, but its role in atherosclerosis remains controversial [22].” But on line 480, the authors then write, “Increased number of senescent Th1 and Th17 CD4+ helper T-lymphocytes were found at the carotid atherosclerotic plaque site secreting high amounts of IL-17, IL-21, IL-23 that accelerates the plaque progression [48].” It would seem that these two points are contradictory.
We would like to apologies for these confusing points. We have decided to remove the line 147 for better understanding.
2. On line 762, “A study proved that senescent macrophages have impaired…” Studies can only support or refute, but they cannot prove.
We are deeply apologies for this grammatical error. We have amended accordingly:
- Original: “A study proved that senescent macrophages have impaired cholesterol efflux …………………… plaque progression”
- Amended: “A study has supported proved that senescent macrophages have impaired cholesterol efflux …………………… plaque progression”
3. There multiple instances of incorrect verb tense used such as on line 111: “Likewise, dendritic cells were the most dominant antigen-presenting cells in cell-mediated immune response by upregulation of cell-surface receptors that act as co-receptors in naïve T cells activation with respective co-stimulatory molecules, major histocompatibility complex MHC I or II.” It should read, “dendritic cells are”, and calling DCs the most dominant is not correct. I would think that the authors mean that DCs are the most potent APCs.
We are deeply apologies for this grammatical error. We have amended accordingly:
- Original: “Likewise, dendritic cells were the most dominant antigen-presenting cells in cell-mediated immune response by upregulation of cell-surface receptors that act as co-receptors in naïve T cells activation with respective co-stimulatory molecules, major histocompatibility complex MHC I or II. OxLDLs are turned into foam cells through ex-ophagy by mature dendritic cells, where the substances are too large to be internalized by phagocytosis.”
- Amended: “Dendritic cells (DCs) are the most potent antigen-presenting cells (APCs) in cell-mediated immune response through upregulation of cell surface receptors that act as co-receptors in naïve T cells activation with respective co-stimulatory molecules including major histocompatibility complex MHC I or II. Extensively oxLDLs are turned into foam cells through exophagy by mature DCs lead to atherosclerotic plaque formation.”
4. There are inconsistencies in the use of singular and plural nouns such as on line 212: “characterized by increased secretion of proinflammatory mediators from macrophage, dendritic cells, and foam cells that aggravates the atherosclerotic plaque formation [51].” Macrophage should be plural.
We are deeply apologies for this grammatical error. We have revised entire manuscript and amended accordingly to ensure the consistency in the use of singular and plural nouns.
- Original: “Senescent immune cells have been found in the vasculature wall, characterized by in-creased secretion of proinflammatory mediators from macrophage, dendritic cells, and foam cells that aggravates the atherosclerotic plaque formation.”
- Amended: “Senescent immune cells have been found in the vasculature wall, characterized by in-creased secretion of proinflammatory mediators from macrophages, DCs, and foam cells that aggravate the atherosclerotic plaque formation”
5. There are still numerous instances of awkward subject/object-pronoun disagreement, such as this on line 825: “The involvement of senescent macrophages in atherosclerosis was further supported when it can secrete….”
We are deeply apologies for this mistake. We have revised accordingly:
- Original: “The involvement of senescent macrophages in atherosclerosis was further supported when it can secrete MMPs to promote plaque instability, elastic fiber degradation and fi-brous cap thinning, therefore, accelerates the lesion rupture and thrombosis [51,139, 152].”
- Amended: “The involvement of senescent macrophages in atherosclerosis is affirmed by its role in secreting MMPs to promote plaque instability, elastic fiber degradation and fibrous cap thinning, therefore, accelerate the lesion rupture and thrombosis [51,139, 152]”
6. There are still numerous instances of incorrect English language usage, such as line 907: “Senescent T-cells have shown to activate p38… “ I suggest something like “have been shown to have activated”… which sounds more appropriate.
We are deeply apologies for this and thank you for the suggestion. With this, we have amended accordingly:
- Original: “Senescent T-cells have shown to activate p38 mitogen-activated protein kinase (p38-MAPK) signaling pathway and expresses IL-18, IFN-γ and CCR-7 that increases the abundance of CD4+/CD8+ (TEMRA) cells.”
- Amended: “Senescent T-cells have been shown to have activated p38 mitogen-activated protein kinase (p38-MAPK) signaling pathway and expressed IL-18, IFN-γ and CCR-7 that in turn increased the abundance of CD4+/CD8+ (TEMRA) cells.”
Reviewer 2 Report
The clarity and importance of the manuscript is much improved. Unnecessary information has been removed and the manuscript shortened.
Author Response
The clarity and importance of the manuscript is much improved. Unnecessary information has been removed and the manuscript shortened.
We would like to thank you for your kind suggestions and comments previously. The current manuscript version has been improved for better clarity and understanding.